# Does the presence of health insurance and health facilities improve access to healthcare for major morbidities among Indigenous communities and older widows in India? Evidence from India Human Development Surveys I and II

**Mathew Sunil George ⓘ \*, Theo Niyosenga, Itismita Mohanty**

Health Research Institute, University of Canberra, Canberra, ACT, Australia

\* sunil.george@canberra.edu.au

**Data Availability Statement:** Data used in this manuscript is publicly available and can be

## Abstract

In this paper, we examine whether access to treatment for major morbidity conditions is determined by the social class of the person who needs treatment. Secondly, we assess whether health insurance coverage and the presence of a PHC have any significant impact on the utilisation of health services, either public or private, for treatment and, more importantly, whether the presence of health insurance and PHC modify the treatment use behaviour for the two excluded communities of interest namely Indigenous communities and older widows using data from two rounds (2005 and 2012) of the nationally representative India Human Development Survey (IHDS). We estimated a multilevel mixed effects model with treatment for major morbidity as the outcome variable and social groups, older widows, the presence of a PHC and the survey wave as the main explanatory variables. The results confirmed access to treatment for major morbidity was affected by social class with Indigenous communities and older widows less likely to access treatment. Health insurance coverage did not have an effect that was large enough to induce a positive change in the likelihood of accessing treatment. The presence of a functional PHC increased the likelihood of treatment for all social groups except Indigenous communities. This is not surprising as Indigenous communities generally live in locations where the terrain is more challenging and decentralised healthcare up to the PHC might not work as effectively as it does for others. The social class to which one belongs has a significant impact on the ability of a person to access healthcare. Efforts to address inequity needs to take this into account and design interventions that are decentralised and planned with the involvement of local communities to be effective. Merely addressing one or two barriers to access in an isolated fashion will not lead to equitable access.

accessed from the IHDS database at the following link https://ihds.umd.edu.

**Funding:** The authors received no specific funding for this work.

**Competing interests:** The authors have declared that no competing interests exist.

## Background

Inequitable access to healthcare is a critical issue that affects health systems across the world [1–5]. Despite advances in healthcare technologies, nearly half the world's population continues to struggle to access basic healthcare services [6]. Recognising the problem of inequitable access to healthcare, the World Health Organization (WHO) has proposed universal health coverage (UHC) as a key policy to enable equitable access to health services across the world [7]. The WHO defines UHC as 'ensuring that all people can use the promotive, preventive, curative, rehabilitative and palliative health services they need, of sufficient quality to be effective, while also ensuring that the use of these services does not expose the user to financial hardship'. Though the key principles of UHC are defined by the WHO, each country is given the freedom to evolve what is UHC for its national health system, considering the contextual factors that are responsible for inequitable access in the first place. However, most UHC programmes, in practice, place a strong emphasis on addressing the financial barriers to accessing healthcare services by covering at least part of the direct costs at the point of care [8]. Health insurance schemes that address direct costs associated with a predefined package of health services delivered through various health facilities have become a common feature of UHC schemes, especially in the developing world [8]. By reducing costs associated with accessing healthcare, it is hoped that, over time, such schemes will also encourage greater use of health services [9, 10]. Nevertheless, most insurance based UHC schemes only address direct payments at a healthcare facility for a selected set of health conditions. They do not cover health expenditure that is not listed in the scheme, nor any kind of outpatient expenditure. Moreover, other expenses incurred–such as transport costs, loss of wages for the caregiver, income lost due to illness etc.–are typically not covered by UHC schemes [11, 12].

In India, the Rastra Swasthya Bima Yojana (RSBY) scheme, rolled out in 2008, was the first pan-Indian health insurance scheme [13]. RSBY sought to address high levels of out-of-pocket payments, considered key barriers to accessing healthcare among the poor in India [8, 14, 15]. Eligibility for RSBY coverage was initially set for all below-poverty-line (BPL) groups, with the government providing a full premium subsidy–BPL being defined by the government as families with an annual income below INR 27,000 [16]. Eligibility was later extended to certain groups beyond the BPL category [17]. RSBY provided financial protection up to INR 30,000 on a family floater basis per annum. A family floater policy is a health insurance plan that covers the entire family on the payment of a single annual premium. The sum assured covers the entire family and can be used in case of multiple hospitalisations in the family. States that already had more generous health insurance schemes continued with them, while RSBY covered those states–or in some cases, districts within a state–that were not covered by similar health insurance programmes [18]. In 2018, a revised national health insurance scheme called the Prime Minister's Jan Arogya Yojana (PMJAY) was launched as a follow-up to the RSBY scheme, with RSBY ultimately subsumed under the new scheme. The new scheme provides a coverage of INR 500,000 on a family floater basis to each family enrolled in the scheme for conditions that require admission to a health facility; no outpatient visits are covered and admission to a hospital is covered only for care packages that are approved and post 24 hours of admission. Any condition that requires treatment without admission and is not on the list of approved packages of treatment is not covered. PMJAY was considered to be a key step towards the achievement of UHC in India. At present, the plan to achieve UHC in India comprises two schemes–the PMJAY insurance (also known as the National Health Protection Mission) and Health and wellness centres. Health and wellness centres are envisioned to provide comprehensive primary care and involve upgrading existing Primary Health Care (PHC) facilities, whereas the National Health Protection Mission (NHPM) provides financial risk protection arising out of hospitalisation [19].

While health systems in both developed and developing countries report inequity in access to healthcare, a common feature of those who suffer from such differential access to treatment is that they belong to socially excluded communities [20–22]. Social exclusion of certain groups within a society is a universal phenomenon. Such groups include Indigenous communities, older adults, people belonging to certain ethnic/religious groups or living with specific disease conditions including HIV/AIDS and others. Social exclusion is a dynamic process that is driven by unequal power relationships and works at various levels [23]. Social exclusion has impacts on health at two levels. First, it impacts the ability of excluded communities to access primary resources (including adequate nutrition, education, housing etc.), which in turn directly affects their health status. Secondly, it creates barriers to access rooted in the very factors that led to exclusion, such as caste, sexual orientation, disease status etc., resulting in inequitable access to healthcare services [24]. Inequity in access to healthcare is of serious concern as many of those who suffer from poor access to healthcare also suffer from poor health to begin with. Thus, inequity to access leads to greater deprivation for these individuals and communities.

Social exclusion of certain groups has been extensively documented in India, especially through the lens of the caste system. The caste system is a 3000-year-old model of social stratification that originated in occupation-based categories and that divides the population into mutually exclusive, exhaustive, heredity-based endogamous categories with a clear hierarchy [25]. The caste of an individual is crucial in determining outcomes in almost all spheres of life in India [26]. Exclusionary processes associated with the caste system lead to unequal access to important resources and, therefore, to unfavourable outcomes for those who are lower in the caste hierarchy [27]. The pervasive nature of the caste system can be seen in the fact that even in a progressive state in India, such as Kerala, the caste system remains a cause of societal inequalities [25]. The contributions of the caste system to poor health status and inadequate access to resources have been thoroughly documented and critiqued [28–32]. Those who are lower in the caste categories typically face poorer health outcomes. In addition to poor health, they also face reduced access to healthcare compared to those higher up the caste system [26, 33, 34].

In this paper, we examine the differentials in healthcare access for two socially excluded communities in India—Indigenous communities who are historically socially excluded and older widows who are an emerging community given the demographic profile of India. Indigenous communities, referred to as Adivasis in India, are not considered to be part of the traditional caste structure in India, which is based on Hinduism. However, their socioeconomic conditions place them at the bottom of the development ladder. When it comes to their health status and access to services, Indigenous communities in India mirror other First Nations people across the globe. On typical indicators of health–such as life expectancy, infant mortality, maternal mortality, and other indicators of maternal and child health–Indigenous communities are far behind other social groups in India [35, 36]. Likewise, social exclusion based on gender and ageing is also widely prevalent and accepted as responsible for poor access to resources [37–40]. Further, marital status, particularly widowhood, can lead to social stigma and has been found to be associated with poor health and healthcare utilisation among older adults in other settings [41]. Several studies have pointed to worse health outcomes associated with older widows in India as opposed to their married counterparts [42–46].

While there has been much research to understand the impact of RSBY on access to healthcare, very little of this research has focused on the specific excluded communities that are targeted in the current study–the Indigenous communities and widows aged 60 years and above. Additionally, many of the studies looking at the impact of RSBY and health insurance have focused on healthcare expenditure, out-of-pocket expenditure and maternal and child health-

related usage [14, 47, 48]. The India Human Development Survey (IHDS) has the unique advantage of being the only panel data available in India, with the first wave of data collected in 2005 prior to the implementation of RSBY and the second wave collected post the roll-out in 2012 [49]. This dataset offers a unique opportunity to study the impact of health insurance on access to treatment for those who belong to different social groups, in a way that other well-known datasets in India such as the National Family Health Surveys and the National Sample Survey Organisation's datasets cannot support.

This paper forms the quantitative component of a larger mixed methods study that looked at the issue of equitable access to healthcare from the perspective of two excluded communities, Indigenous and widows above the age of 60 (hereafter referred to in this paper as widows), in the south Indian state of Kerala. The qualitative results from this study have already been published and showed how social exclusion played an important role in creating barriers to equitable access for both communities despite the presence of financial protection schemes including health insurance coverage [50, 51].

This paper therefore has two specific aims. First, we seek to understand whether access to treatment for major morbidity conditions is determined by the social class of the person who needs treatment. Second, we seek to investigate whether health insurance coverage and the presence of a PHC have any significant impact on the utilisation of health services, either public or private, for treatment and, more importantly, whether the presence of health insurance and PHC modify the treatment use behaviour for the two excluded communities of interest (Indigenous people and widows).

## Methods

### Data source and sample

The India Human Development Survey (IHDS) is a nationally representative survey that contains a wide range of information on health, education, employment, economic and marital status, social capital, infrastructure, etc [49, 52]. The first survey round, IHDS1 (2004–2005), involved 41,554 households in 1503 villages and 971 urban areas across India. The second wave, IHDS II (2011–2012), included about 83% of wave 1 households. Additional households were added to wave 2 to maintain a representative sample (n = 42,152). The large sample size of IHDS waves allows us to analyse different segments of the population for which data are available as well as offering the advantages of power and precision [49]. For this study, we used both waves of the IHDS and the information available through the household and village level questionnaires. The household questionnaire contains information on occurrence and duration of short-term and major morbidities, treatment received including hospitalisation, costs incurred, highest educational status of adults in the household, monthly consumption per capita and health insurance coverage. The village questionnaire has information on a wide range of topics, including the presence of various types of public and private healthcare facilities that the community can access for their healthcare needs [34, 53].

IHDS provides data on two types of morbidity–short term and major. Short-term morbidity was assessed by asking the question, 'Has anybody been ill with fever, cough or diarrhoea in the last month?'. Major morbidity was assessed for a set of long-term conditions including cataract, tuberculosis, hypertension, heart disease, diabetes, cancers, leprosy, asthma, polio, paralysis, epilepsy, mental illness, STD or AIDS and any other long-term conditions. The question asked was whether anyone in the household was diagnosed by a doctor with any of the conditions above and if they had received any treatment or advice in the last 12 months preceding the survey.

Our sample includes individuals with any of the health conditions categorised as major morbidities available in both waves in 33 states and 68 districts across India. The sample

consists of a total of 34,604 individuals in the IHDS dataset for whom we have information on any long-term health condition in both waves. Records with missing values were dropped during data cleaning and a final sample of 34,587 records was used for the analysis.

The data are clustered and hierarchically structured, with individuals nested within households, then households nested within communities, communities within villages, villages within districts and, finally, districts within states. Individuals who are suffering from illness within the same household are more likely to have similar outcomes with regard to treatment access than others chosen randomly from the population at large. Households located within the same community are more likely to face the same set of cultural and structural facilitators or barriers when it comes to access to treatment. Similarly, families located within a particular village or neighbourhood are more likely to share similar infrastructure, including the presence of health facilities where treatment can be accessed [54]. Villages and urban neighbourhoods are located within administrative units such as districts and states. At the policy level, healthcare is a matter for state-level governments in India; therefore, the policies of each state government have an impact on several critical aspects of the provision and availability of health services, including infrastructure. Various aspects of the social determinants of health, including access to education, are also influenced by each state government and its policies. However, for these analyses we will consider three levels of hierarchy: individuals, households, and states.

The qualitative data that is referred to in this paper was carried out among Indigenous communities and older widows in the south Indian state of Kerala. Ethnographic fieldwork was conducted by the first author who is fluent in the native language of the participants (Malayalam/Tamil) and has prior experience working with marginalised Indian communities. Among the Indigenous communities living in Attapadi (Attapadi is a tribal block located in Mannarghat taluk in the Palakkad district of Kerala, comprising 192 villages inhabited by members of the muduga, the kurumba and the irula Indigenous communities), this occurred between August 2018 and January 2019 and again between August 2019 and October 2019. Among the older widows living by themselves in Kottayam district, the fieldwork was carried out between November 2018 and January 2019 and again between October 2019 and January 2020. Details regarding the methods, analysis and key findings from the qualitative arm have already been published [50, 51].

Written informed consent was gained from individual participants prior to data collection. In situations where written informed consent was not possible to obtain, the informed consent process was witnessed by the village chief (in the case of the Indigenous communities) or a health worker (in case of the older widows). The Human Research Ethics Committees of the University of Canberra [20180074] and the Indian Institute of Public Health Delhi [IIPH-D_IEC_ 03_2018] provided ethical approval. Regulatory permissions were obtained from the Kerala Department of Health [GO(4)No2677/2018/H&FWD], the local administration in Attapadi as well as the National Health Mission office at Kottayam.

### Description of variables

**Outcome variable.** The outcome variable is a binary choice variable, defined as '1' if a person with major morbidity has received treatment for any of the conditions listed under major morbidity and '0' if they have not. This is in response to the question in the two waves of IHDS, 'In the last twelve months, have you (person with a major morbidity) received any treatment or advice?'. In this study, we look only at access to treatment for major morbidity, as the three conditions listed under short-term conditions are more likely to be treated using home remedies or non-prescription medication than in a formal healthcare setting. We define access

to treatment as the ability of an individual who required healthcare to obtain it irrespective of their social class.

**Explanatory variables.** The explanatory variables–such social class, widowhood, health insurance and presence of a PHC–are grouped into two levels reflecting the hierarchical nature of the data. At the individual level are the social class of the participants, whether they are widows above the age of 60 and health insurance coverage status. At the village/neighbourhood level is the presence of a PHC. We standardised social class across the two waves into five different categories based on IHDS data. Thus, *brahmins* and other higher castes have been combined to form a category called 'higher caste', followed by other backward castes, Dalits, Adivasi*s*, Muslims and, finally, Christians, Sikhs and Jains as a single category. The proportions of different caste groups reported by IHDS are similar to those reported by others and representative of the Indian population [Baru et al., 2010]. In order to understand the ability of widows above the age of 60 to access treatment, we created a variable to capture widows above the age of 60 in our sample from both waves using the variables of age, gender and marital status, available in IHDS.

**Control variables.** We controlled for income and education since one of the explanatory variables of interest–social class–has been shown to be positively associated with both education and income [25, 55]. Education is proxied by the highest education of the adult in the family (defined as no formal education, primary, middle school, secondary, higher secondary, and graduate and above). We controlled for income using monthly consumption per capita per household divided into quintiles. Place of residence (rural vs urban) was also controlled for to account for any differences in healthcare provision and constraints on either the demand or supply side due to location. We also created a new variable, Region, based on the six official regional classifications of states in India [56]. This allowed us to control for the variations across the different regions in India. This is important since policies and programmes related to healthcare vary considerably across India.

**Summary statistics.** Table 1 (below) presents the descriptive statistics of the analysed sample of those suffering from major morbidity and requiring treatment. Our sample contains substantially more people with major morbidities in wave 2 (n = 21789) than in wave 1 of IHDS (n = 12,798). Wave 1 seeks details about 14 long-term health conditions. Wave 2 collects details on the 14 conditions listed in wave 1 and adds one more condition, accidents. Other backward castes (OBCs) form the highest proportion of the sample, followed by those who belong to higher castes. The proportion of the Indigenous communities in the original sample is 8.28 per cent, which is in line with the proportion of Adivasis in India. However, since we exclude all those who do not have any of the major morbidities listed in the IHDS dataset, the proportion of the Indigenous communities falls to 4.40 per cent in our final sample. Only 8.63 per cent of the sample is covered by health insurance, either public or private. Health facilities, measured by the presence of a PHC, are found across 48.66 per cent of the sample.

## Statistical analyses

Multi-level mixed effect models were used to account for the clustered nature of the data [54]. Initially, a null model was fitted, in which only state and household levels random effects are accounted for, in order to estimate the magnitude of variability due to households and state levels (Table 2). A series of multi-level mixed effect logistic regression models were run, with treatment for major morbidity as the outcome variable, and social groups (five groups with the high caste group as the reference group), widows aged 60 or above, presence of health facility (PHC) and survey wave (2011–2012 vs 2004–2005) as the main explanatory variables and then added the control variables into the final model (Table 3). Model 0 (Table 3) presents the

**Table 1. Descriptive statistics.**

| Variable n = 34,587 | Proportion (%) or Mean | Standard Deviation | Minimum value | Maximum value |
|---|---|---|---|---|
| Obtained treatment | | | | |
| Wave 1 (n = 11,776) | 91.94% | | 0 | 1 |
| Wave 2 (n = 20,014) | 91.76% | | | |
| **Social Groups** | | | | |
| Higher Caste | 24.94 | 0.43 | 0 | 1 |
| Other backward caste | 34.20 | 0.45 | 0 | 1 |
| Dalits | 18.90 | 0.39 | 0 | 1 |
| Adivasis | 4.40 | 0.21 | 0 | 1 |
| Muslims | 12.66 | 0.33 | 0 | 1 |
| Christians, Sikhs, Jains | 4.88 | 0.21 | 0 | 1 |
| Widows>60 | 8.63 | 0.28 | 0 | 1 |
| Insurance | 8.63 | 0.29 | 0 | 1 |
| Presence of a PHC | 48.66 | 0.49 | 0 | 1 |
| **Highest Education of adult in the family** | | | | |
| No formal education | 14.49 | 0.35 | 0 | 1 |
| Primary (1–5 yrs.) | 12.50 | 0.33 | 0 | 1 |
| Middle (6–8 yrs.) | 14.09 | 0.34 | 0 | 1 |
| Secondary (9–10 yrs.) | 21.63 | 0.41 | 0 | 1 |
| Diploma/Higher Secondary (11–14 yrs.) | 16.64 | 0.37 | 0 | 1 |
| Graduate & Above (15 plus yrs.) | 20.66 | 0.40 | 0 | 1 |
| Rural/Urban | 38.4 | 0.48 | 0 | 1 |
| **Region** | | | | |
| North | 22.27 | 0.42 | 0 | 1 |
| North East | 1.74 | 0.13 | 0 | 1 |
| Central | 22.99 | 0.41 | 0 | 1 |
| East | 17.73 | 0.38 | 0 | 1 |
| West | 10.68 | 0.31 | 0 | 1 |
| South | 25.58 | 0.44 | 0 | 1 |

simplest bivariate associations between access to treatment and each individual factor of interest. Model 1 is the simplest multivariable model showing adjusted effects (as opposed to crude effects in Model 0), with social groups and widows as the main effects of interest since they represent community exclusion. Both community exclusion groups as main effects and their interactions with health insurance, primary healthcare facility and survey wave were examined in Model 2 (Table 3). Then, in the final model (Model 3), control variables age, education, monthly household consumption per capita, location and regional differences were added as covariates.

In all model results, regression coefficients are presented in the log-odds scale. They represent differences in the log-odds scale; they have not been expressed. In the odds ratio scale, as

**Table 2. Null model–model with intercept only and random effects (state and household).**

| Random effects | Variance component (95% CI) | ICC (%) (95% CI) |
|---|---|---|
| Between Household variance | 4.896 (3.76; 6.38) | 55.99 (51.15; 59.63) |
| Between State variance | 0.558 (0.301; 1.03) | 6.40 (3.58; 11.13) |

Notes: ICC–Intraclass correlation (percentage of variance at the level of hierarchy)

**Table 3. Multilevel mixed effects models of treatment seeking.**

| No of observations: N = 34,587 | Model 0: Binary Models of Individual factors only: Coeff. (95% C.I) | Model 1 | Model 2: Main effects, interactions groups with insurance, PHC, survey wave: Coeff. (95% CI) | Model 3: Model 2 with adjustment for covariates: Coeff. (95% CI) |
|---|---|---|---|---|
| **Main effects:** | | | | |
| OBC | −0.373*** (−0.540; −0.205) | −0.293* (−0.459; −0.127) | −0.626*** (−0.936; −0.317) | −0.372* (−0.681; −0.063) |
| Dalits | −0.520*** (−0.701; −0.339) | −0.430*** (−0.609; −0.251) | −0.530* (−0.873; −0.186) | −0.094 (−0.441; 0.253) |
| Adivasis | −1.12*** (−1.39; −0.853) | −0.979*** (−1.24; −0.714) | −1.515*** (−1.97; −1.06) | −0.957*** (−1.413; −0.500) |
| Muslims | 0.121 (−0.109; 0.351) | 0.134 (−0.094; 0.361) | −0.278 (−0.711; 0.155) | 0.107 (−0.327; 0.542) |
| Christians, Sikhs, Jains | −0.151 (−0.522; 0.219) | −0.154 (−0.519; 0.211) | 0.400 (−0.371; 1.172) | 0.228 (−0.533; 0.989) |
| Widows>60 | −0.436*** (−0.620; −0.251) | −0.464*** (−0.646; −0.282) | −0.664*** (−1.030; −0.297) | −0.800*** (−1.186; −4.137) |
| Health Insurance | 0.125 (−0.085; 0.334) | 0.075 (−0.133; 0.282) | 0.490* (0.044; 0.936) | 0.323 (−.117; 0.763) |
| Health Facility (PHC) | 0.636*** (0.504; 0.769) | 0.545*** (0.414; 0.675) | 0.572*** (0.315; 0.829) | 0.067 (−0.232; 0.367) |
| Survey Wave (2nd) | 0.033 (−0.091; 0.158) | 0.037 (−0.086; 0.161) | −0.412* (−0.679; −0.144) | −2.31*** (−2.994; −1.635) |
| **Interactions:** | | | | |
| Social groups # Insurance | | | | |
| OBC # insurance | | | −0.733* (−1.28; −0.184) | −0.658* (−1.200; −0.116) |
| Dalits # insurance | | | −0.515 (−1.14; .108) | −0.459 (−1.075; 0.158) |
| Adivasis # insurance | | | 0.269 (−.749; 1.287) | 0.380 (−0.629; 1.389) |
| Muslims # insurance | | | −0.708 (−1.568; 0.153) | −0.608 (−1.459; 0.243) |
| Christians Sikhs Jains # insurance | | | −0.274 (−1.301; 0.753) | −0.114 (−1.135; 0.907) |
| Widows # insurance | | | 0.163 (−0.435; 0.761) | 0.151 (−0.443; 0.746) |
| Social groups # PHC | | | | |
| OBC # PHC | | | −0.120 (−0.443; 0.202) | −0.015 (−0.335; 0.305) |
| Dalits # PHC | | | 0.057 (−0.309; 0.424) | 0.173 (−0.193; 0.538) |
| Adivasis # PHC | | | 0.455 (−0.146; 1.06) | 0.485 (−0.113; 1.08) |
| Muslims # PHC | | | −0.255 (−0.703; 0.194) | 1.161 (−0.608; 0.286) |
| Christians, Sikhs, Jains # PHC | | | −0.786* (−1.56; −0.15) | −0.545 (−1.31; 0.215) |
| Widows # PHC | | | 0.265 (−0.1035; 0.633) | 0.312 (−0.443; 0.746) |
| Survey # social groups | | | | |
| Survey # OBCs | | | 0.720*** (0.385; 1.05) | 0.573** (0.236; 0.909) |
| Survey # Dalits | | | 0.212 (−0.164; 0.590) | 0.035 (−0.349; 0.419) |
| Survey # Adivasis | | | 0.685* (0.152; 1.22) | 0.457 (−0.085; 0.999) |
| Survey # Muslims | | | 0.944*** (0.479; 1.41) | 0.815** (0.349; 1.28) |
| Survey # Christian, Sikh, Jain | | | −0.064 (−0.782; 0.654) | −0.048 (−0.760; 0.663) |
| Survey # Widows | | | 0.087 (−0.311; 0.484) | 0.111 (−0.286; 0.507) |
| **Covariates:** | | | | |
| *Region* | | | | |
| North | | | | −0.537 (−1.23; 0.153) |
| North−east | | | | −0.478 (−1.37; 0.411) |

(*Continued*)

**Table 3.** (Continued)

| No of observations: N = 34,587 | Model 0: Binary Models of Individual factors only: Coeff. (95% C.I) | Model 1 | Model 2: Main effects, interactions groups with insurance, PHC, survey wave: Coeff. (95% CI) | Model 3: Model 2 with adjustment for covariates: Coeff. (95% CI) |
|---|---|---|---|---|
| Central | | | | −0.789 * (−1.55; −0.028) |
| East | | | | −1.05** (−1.80; −0.292) |
| West | | | | −0.506 (−1.329; 0.317) |
| *COPC* | | | | |
| 2nd quartile | | | | 0.818*** (0.608; 1.03) |
| 3rd quartile | | | | 2.16*** (1.50; 2.83) |
| 4th quartile | | | | 2.74*** (2.05; 3.43) |
| *Highest education of adult in the family* | | | | |
| Primary (1−5 yrs education) | | | | 0.215* (0.011; 0.418) |
| Middle (6−8 years education) | | | | 0.238* (0.036; 0.439) |
| Secondary (9−10 yrs education) | | | | 0.436*** (0.240; 0.631) |
| Diploma/ higher sec (11 −14 years education) | | | | 0.458*** (0.243; 0.673) |
| Graduate & above | | | | 0.643*** (0.416; 0.870) |
| *Location* (Urban) | | | | 0.225* (0.018; 0.432) |
| Age groups | | | | |
| 16−30 | | | | −0.423*** (−0.682; 1.164) |
| 31−45 | | | | −0.056 (−0.297; 0.185) |
| 46−60 | | | | 0.102 (−0.138; 0.341) |
| 60+ | | | | 0.137 (−0.118; 0.392) |
| Variance components | | | | |
| | Between Households | 4.41 (3.50; 5.56) | 4.44 (3.525; 5.582) | 4.18 (3.355; 5.207) |
| | Between States | 0.49 (0.260; 0.914) | 0.49 (0.261; 0.914) | 0.295 (0.153; 0.570) |

**Notes:** All p−values are replaced by stars (with

***: $p < 0.0001$

**: $p < 0.001$

*: $p < 0.05$)

OBC: Other Backward Castes; PHC: Primary health centre; COPC: Monthly per capita consumption

Reference groups

Waves: Wave 1; Social groups: Higher caste; Widows, Insurance, Presence of PHC: No; Region: South; Highest education of adult in the family: No formal education; Location: Rural; COPC quartiles: First quartile; Age groups: 0–15 years.

it is often the case for Binary outcome variable and logistic regression models. This presentation of results in the additive log-odds scale allows easy interpretation of the coefficients associated to the interaction terms which represent the moderation effects.

## Results

We initially estimated a null model with only the state and household levels of random effects (Table 2). The intraclass correlation coefficient (ICC) indicates that 56.0% per cent of the total variation in access to treatment for major morbidity comes from households, while only 6.40

per cent comes from the variation between the states. The remaining 37.6 per cent is potentially due to individual-level variations in seeking treatment for major morbidity.

## Social exclusion and its impact on access to treatment

Both unadjusted and adjusted models' findings highlight the negative effect of social exclusion on the access to treatment (Model 0 and Model 1, Table 3). Model 1 (Table 3) shows that, compared to those who belong to higher castes, Dalits and the Indigenous communities in India have statistically significant adjusted negative effects (i.e., are disadvantaged), meaning that they were less likely to access treatment for major morbid conditions. Widows were also seen to be significantly disadvantaged (negative effect) when it came to accessing treatment for major morbidity. Among these communities, the Indigenous communities in India (Adivasis) were by far the most disadvantaged, followed by Dalits.

We looked at the simultaneous moderation effects of insurance, presence of health facility (PHC) and survey wave by interacting each with social groups (Model 2, Table 3). With regards to insurance, only OBCs exhibited a statistically significant negative differential score, all other social classes' differential scores being statistically non-significant. For the presence of a PHC, the Indigenous communities recorded a positive but statistically non-significant effect modification, whereas Christians, Sikhs and Jains had a negative and statistically significant differential effect. However, none of these differential scores was large enough to affect the direction of the adjusted scores (the impacts) of these social exclusion groups. As for survey wave, estimated moderation effects were statistically significant and positive for Muslims, OBCs and Indigenous communities (Model 2, Table 3). For the OBCs, the positive differential score was large enough to turn the effect on access to treatment from negative (in the first wave) to positive effect in the second wave, although the size of the effect is non-significant. Thus, except for the OBCs and Christians, Sikhs and Jains, other social classes continued to experience poor access to treatment, indicating that the disadvantages experienced by the community were not overcome by any changes that had occurred between the two survey waves.

When we controlled for potential confounders (Model 3, Table 3), the Indigenous communities and the widows in the sample continued to show a negative and statistically significant impact scores (main effects which are effects observed in the first survey round, in the absence of insurance and PHC). As shown in Table 3, controlling for covariates did affect slightly the magnitude of differential effects but not their statistical significance. In some cases, the magnitude of effects dropped due to adjustment for covariates, suggesting some of these covariates–such as the highest education of the adult in the family and the monthly consumption per capita–appear to be stronger predictors of access to treatment than social groups.

## Effect of health insurance

Health insurance coverage showed, overall, a marginally statistically non-significant positive effect with regard to accessing treatment for major morbidity (Table 3, Models 0, 1). Marginally statistically non-significant variables are those which are approaching significance. Reporting these variables is important in the overall context of this paper. In all models including interactions with the different social groups (Models 2, 3 in Table 3), we observed a positive effect modification for insurance, Indigenous communities and widows, although statistically non-significant and not large enough to induce a positive change in the likelihood of accessing treatment.

## Presence of health facilities

The availability of a PHC showed an overall statistically significant and positive association with access to treatment (Table 3, Models 0, 1), meaning that the PHC increased the likelihood

of seeking treatment. Further interaction analyses of PHC with the different social groups and widows showed that PHC increased the likelihood of treatment (a positive effect modification) for Indigenous communities and widows, but that this was statistically non-significant, whereas for religion-based group (Christians, Sikhs and Jains), the presence of a PHC showed a negative effect modification that was statistically significant (Model 2 in Table 3). However, these effect modification scores were not large enough to alter the adjusted effects of these social groups on the probability of access to treatment. When we looked at the main effect as well as the interactions between the social groups, widows and health insurance and PHC, the presence of the PHC continued to show a positive effect with strong statistical significance (Table 3, Model 2). However, controlling for confounders altered the effect of the presence of a PHC on treatment for major morbidity which, although still positive, was no longer statistically significant (Table 3, Model 3).

Among the covariates controlled for in the final model (Model 3, Table 3), the levels of the household education and the consumption per capita were positively associated with access to treatment, with the magnitude of effects increasing with increased levels of education (primary to graduate and above, compared to no formal education) and quartiles of consumption (2nd to 4th quartile, compared to first quartile). Urban residency was positively associated with access to treatment, while for age, only the young age group showed a statistically significant negative effect on health utilisation compared with all the rest of the age groups. As for regions, in comparison with the southern region, all the rest had negative effects on accessibility to treatment, but this was large in magnitude and statistically significant only for the Central and Eastern regions. Thus, in comparison to other parts of India, the southern states are doing better in terms of accessibility to treatment.

## Discussion

The impact of social exclusion on access to various resources, including healthcare, is well established across various parts of the world [23, 37]. The global movement towards UHC has tried to address the inequity in access to healthcare by proposing an approach that focuses on financial protection for those who need healthcare services and on the quality and coverage of services [7]. Most interventions that have been introduced to achieve UHC, especially in developing countries, have taken the form of health insurance schemes [8]. India has designated two schemes–the national health protection scheme, which provides health insurance coverage, and the upgrading of primary health care (PHC) facilities as health and wellness centres– as its key interventions to usher in UHC for its population by 2030 [19]. While this integrated scheme is new, and thus was not completely captured in the IHDS data of 2005 and 2012, the elements of both these schemes–health insurance and PHCs–were present.

This study makes a significant contribution to the literature on inequal healthcare utilisation by examining the overall impact of both these interventions across different social groups in India. Moreover, the study focused on these two interventions and how successful they were in ensuring equitable utilisation of healthcare for some excluded communities–Indigenous communities and widows above 60 years of age. Our analysis adds to the body of literature that inequitable access to healthcare services exists in India and that the social group, to which an individual belongs, plays an important role in determining access to treatment. A key means of operationalising inequality in India has been the caste system. It is well established that social exclusion practised through the caste system has a negative impact for individuals belonging to communities considered to be lower in the caste hierarchy. Widows across various castes are also considered to be lower in social status and therefore have a lesser claim on social resources. This means that such individuals, despite having resources such as health

insurance and the presence of health facilities, are unable to access them unlike others who are considered to be higher in the caste hierarchy and do not suffer from social exclusion.

Our findings showed that both Indigenous communities and elderly widows have reduced access to treatment compared to other social groups in India. For both groups, the presence of health coverage and health facilities did not significantly modify this accessibility issue. The marginal improvement observed (moderation effect) was not large enough to soothe the inequal healthcare accessibility. This has implications for the way in which health service delivery is planned and provisioned, as it strongly suggests that a one-size-fits-all approach to addressing inequitable access will not work. The social contexts of these communities differ considerably from those of other groups. If they are to be effective, interventions to improve access to treatment need to be sensitive to the macro-level determinants of these communities, such as social status, living conditions, literacy, and access to land and nutrition [50, 51, 57].

The qualitative findings of our analysis also demonstrated that health insurance did not have any significant impact on access to treatment for either of these excluded communities. Among the Indigenous communities, the lack of culturally safe healthcare services and issues including power differentials between the community and the health system, and the barriers created by social determinants, were major impediments for the community in accessing healthcare services. Free healthcare provided at primary, secondary and tertiary levels was not, on its own, enough to overcome these barriers and ensure that the community members were able to access treatment when they needed it [50, 51].

Among the widows above 60, while health insurance–both state and private–was found to be common, many of them did not make use of it. Most of the conditions for which the community members sought treatment for required outpatient visits to health facilities and the purchase of medications on their own [51]. The health insurance coverage was limited to specific conditions and was only accessible in the case of admission to a health facility that accepted cashless treatment. In effect, health insurance did not address the most common of their treatment requirements. Evaluations of 24 developing countries and their progress in achieving UHC goals have criticised the focus on out-of-pocket payments [58]. Similarly, research on the impact of financial protection in India has shown that the provision of financial protection through health insurance schemes does not improve access to healthcare [59] or even reduce out-of-pocket expenditure [48, 59–61]. Indeed, an earlier study in India showed that, in cases of extreme deprivation, financial protection alone will not be enough to increase the use of healthcare services [62].

In comparison with health insurance coverage, the presence of a functional PHC showed greater association with better access to treatment. A PHC in the Indian public health system is envisaged as a basic health unit providing integrated curative and preventive healthcare, especially to the rural population [63]. Functional PHCs provide decentralised healthcare and enable greater uptake of health services. However, this association was not seen in the case of the Indigenous communities. Indigenous communities in India are mostly located in difficult to reach and hilly terrain. Therefore, while a functional PHC might be a highly decentralised form of service delivery for other social groups, who typically live in areas where travel is easier, among Indigenous communities, decentralisation below the level of the PHC to provide treatment for common illnesses would be important. As part of the qualitative findings, feedback from the Indigenous communities living in Attapadi, Kerala, demonstrated that centralisation of healthcare services was one of the major barriers to access to healthcare [50]. Among the widows above 60, it was found that decentralisation below the PHC level through neighbourhood clinics that provided preventive and curative primary care for common health conditions, along with free medication, were more effective than PHC-level provision of healthcare [51].

Access to healthcare is a complex issue, with multiple determinants on both demand and supply sides. Merely addressing one or two barriers will not solve this issue in an equitable manner. India will need to take an approach that addresses the social context that mediates access to healthcare. Health insurance and functional and decentralised health facilities are an important step in the right direction. However, the pathways to marginalisation faced by excluded communities must be taken into consideration and steps taken to address them. Interventions to improve access to healthcare interventions should be therefore be locally relevant and planned with the involvement of communities to ensure they are tailored to the social reality of communities.

## Supporting information

**S1 Questionnaire. Inclusivity in global research.**
(DOCX)

## Author Contributions

**Conceptualization:** Mathew Sunil George, Theo Niyosenga, Itismita Mohanty.

**Formal analysis:** Mathew Sunil George, Theo Niyosenga, Itismita Mohanty.

**Methodology:** Theo Niyosenga, Itismita Mohanty.

**Supervision:** Theo Niyosenga, Itismita Mohanty.

**Validation:** Itismita Mohanty.

**Writing – original draft:** Mathew Sunil George.

**Writing – review & editing:** Mathew Sunil George, Theo Niyosenga, Itismita Mohanty.

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
