## [Decision Letter · Decision Letter 0]

26 Apr 2022

PONE-D-21-37852Does the presence of health insurance and health facilities improve healthcare utilisation for major morbidities among Indigenous communities and older widows: Evidence from the India Human Development Surveys I and IIPLOS ONE

Dear Dr. George,

Thank you for submitting your manuscript to PLOS ONE. After careful consideration, we feel that it has merit but does not fully meet PLOS ONE’s publication criteria as it currently stands. Therefore, we invite you to submit a revised version of the manuscript that addresses the points raised during the review process.

We look forward to receiving your revised manuscript.

Kind regards,

Hao Xue

Academic Editor

PLOS ONE

Journal Requirements:

“No: The funders had no role in study design, data collection and analysis, decision to publish, or preparation of the manuscript”

Reviewers' comments:

Reviewer's Responses to Questions

**Comments to the Author**

1. Is the manuscript technically sound, and do the data support the conclusions?

Reviewer #1: Partly

2. Has the statistical analysis been performed appropriately and rigorously? 

Reviewer #1: No

3. Have the authors made all data underlying the findings in their manuscript fully available?

Reviewer #1: No

4. Is the manuscript presented in an intelligible fashion and written in standard English?

Reviewer #1: Yes

5. Review Comments to the Author

Reviewer #1: Dear Editor

This is a good manuscript to publish in your valuable journal, but before making a decision, it needs a major revision, especially in the methods.

1- Analyzes do not include questionnaire questions for example: occurrence and duration of short-term and major morbidities, treatment received including hospitalization, costs incurred,…

2- Coverage ratios (CR) can be another option in presenting the results.

3- Major morbidities are unknown

4- The results are not well described and need to be rewritten.

5- In the dissection, of inequality and health coverage (UHC) and its relationship to outcomes be explained.

6. PLOS authors have the option to publish the peer review history of their article (what does this mean?). If published, this will include your full peer review and any attached files.

Reviewer #1: No

---

## [Author Response · Author response to Decision Letter 0]

19 Sep 2022

1. Is the manuscript technically sound, and do the data support the conclusions? Reviewer #1: Partly

Response:

We believe that this manuscript is fully (not partly) technically sound, and that data analyses and results support the discussions and conclusions formulated. As PLOS One requires, this manuscript describes a technically sound piece of scientific research with data that supports the conclusions. All authors have the required technical expertise for this research.

2. Has the statistical analysis been performed appropriately and rigorously? Reviewer #1: No

Response:

We believe that this is unfair evaluation of the manuscript. As stated above, authors have the required expertise in both quantitative and qualitative methods. Indeed, both Dr Mohanty and Dr Niyonsenga are quantitative methodologists, experts in Health Economics and Biostatistics respectively, with well-established and long-standing experience in quantitative modelling methods. They published extensively in high quality journals including Q1 journals such as PLOS One, so are aware of the rigor needed/required in any publication.

3. Have the authors made all data underlying the findings in their manuscript fully available? Reviewer #1: No

Response:

See data availability statement which gives details of the online location where the raw files of the India Human Development Survey I and II are freely available.

Reviewer #1: Dear Editor

This is a good manuscript to publish in your valuable journal, but before making a decision, it needs a major revision, especially in the methods.

1- Analyzes do not include questionnaire questions for example: occurrence and duration of short-term and major morbidities, treatment received including hospitalization, costs incurred, …

2- Coverage ratios (CR) can be another option in presenting the results.

3- Major morbidities are unknown

4- The results are not well described and need to be rewritten.

5- In the dissection, of inequality and health coverage (UHC) and its relationship to outcomes be explained.

Response:

We thank the reviewer for the statement that this is a good manuscript to publish in the PLOS One valuable journals. This is indeed a very good manuscript and each point raised by the reviewer will be addressed below.

1. Analyzes do not include questionnaire questions for example: occurrence and duration of short-term and major morbidities, treatment received including hospitalization, costs incurred, …

Response:

We thank the reviewer for this comment, and all the questions mentioned above are great examples of potential research questions. Although the information on the type of treatment received, short-term and long-term morbidities and the costs, is available in the IHDS survey data, as reported in the manuscript, the main focus of the paper is on access to treatment. We are looking at the treatment seeking behaviour as the outcome variable; we are talking of people not able to reach to the hospital and that the social class factor is playing a significant part in this despite the presence of health insurance and health facilities. We clearly stated that in the manuscript the following (page 8, lines 225-229).

In this study, we look only at access to treatment for major morbidity, as the three conditions listed under short-term conditions are more likely to be treated using home remedies or non-prescription medication than in a formal healthcare setting. We define access to treatment as the ability of an individual who requires healthcare to obtain it irrespective of their social class.

2. Coverage ratios (CR) can be another option in presenting the results.

Response:

We thank the reviewer for this comment, and it would be interesting to consider the insurance coverage ratios. However, we simply wanted to investigate the effect of the launched health insurance as a step towards the universal health coverage in India. The focus of our paper is on access to healthcare for the two socially excluded populations of interest and not primarily on the extent of health coverage of both populations with health insurance. 

However, we admit that a term we have used in the title of the paper might be misleading and cause readers to miss the key focus of the paper. Hence, we are removing the world “utilisation” from the title and revising it as follows.

Does the presence of health insurance and health facilities improve access to healthcare for major morbidities among Indigenous communities and older widows in India?

3. Major morbidities are unknown

Response:

We thank the reviewer for this comment but politely disagree with this comment as the details of the major morbidities are given in methods section where we describe our Data source and sample (page 6, lines 177-180; page 7, lines 181-183).

4. The results are not well described and need to be rewritten.

Response:

We thank the reviewer for this comment but politely disagree with this comment. As we described in the background section of our manuscript, the key focus of our paper is to understand whether the presence of health insurance and health facilities will have an impact on access to healthcare for socially excluded communities in India (Indigenous communities and Widows older than 60 years). We looked at these two measures as the main intervention to achieve Universal Health Coverage in India, that is, the Pradhan Mantri Jan Arogya Yojana (PMJAY) which seeks to improve access to healthcare through the provision of health insurance, and the health and wellness centres which are upgraded Primary Health Centres. Given this context, we would like to humbly state that the presentation of results is centred around the key variables that capture social exclusion, health insurance coverage and availability of a primary healthcare centre.

In the absence of any specific issues that the reviewer has raised with regard to the results section, we believe our current presentation should be adequate. However, we added the following in statistical analyses section to make clear what these results represent (page 12, lines 285-289).

In all model results, regression coefficients are presented in the log-odds scale. They represent differences in the log-odds scale; they have not been expressed in the odds ratio scale, as it is often the case for Binary outcome variable and logistic regression models. This presentation of results in the additive log-odds scale allows easy interpretation of the coefficients associated to the interaction terms which represent the moderation effects.

5. In the discussion section, of inequality and health coverage (UHC) and its relationship to outcomes be explained.

Response:

We would like to thank the reviewer for this comment. In response, we have further added relevant details in the discussion section (page 22, lines 396-403) as follows. 

A key means of operationalising inequality in India has been the caste system. It is well established that social exclusion practised through the caste system has a negative impact for individuals belonging to communities considered to be lower in the caste hierarchy. Widows across various castes are also considered to be lower in social status and therefore have a lesser claim on social resources. This means that such individuals despite having resources such as health insurance and the presence of health facilities are unable to access them unlike others who are considered to be higher in the caste hierarchy and do not suffer from social exclusion. 

We have also added the following sentences to explain the observed relationships (page 22, lines 405-408).

For both groups, the presence of health coverage and health facilities did not significantly modify this accessibility issue. The marginal improvement observed (moderation effect) was not large enough to soothe the inequal healthcare accessibility.

The discussion section made a link between quantitative results and earlier qualitative study findings (page 23 lines 414-421), showing that the observed relationships between access to treatment and the presence of both health coverage and health facilities were corroborated by the qualitative analysis. The qualitative piece of findings offered sound explanations of the negative relationships observed (see also page 23, lines 422-434).

---

## [Editor Report · Decision Letter 1]

26 Jan 2023

Does the presence of health insurance and health facilities improve access to healthcare for major morbidities among Indigenous communities and older widows in India? Evidence from India Human Development Surveys I and II

PONE-D-21-37852R1

Dear Dr. George,

We’re pleased to inform you that your manuscript has been judged scientifically suitable for publication and will be formally accepted for publication once it meets all outstanding technical requirements.

Kind regards,

Kannan Navaneetham, PhD

Academic Editor

PLOS ONE
---

## [Editor Report · Acceptance letter]

30 Jan 2023

PONE-D-21-37852R1 

Does the presence of health insurance and health facilities improve access to healthcare for major morbidities among Indigenous communities and older widows in India? Evidence from India Human Development Surveys I and II 

Dear Dr. George:

I'm pleased to inform you that your manuscript has been deemed suitable for publication in PLOS ONE. Congratulations! Your manuscript is now with our production department. 

Kind regards, 

on behalf of

Prof. Kannan Navaneetham 

Academic Editor

PLOS ONE